# Influence of CT Image Matrix Size and Kernel Type on the Assessment of HRCT in Patients with SSC-ILD

**DOI:** 10.3390/diagnostics12071662

**Published:** 2022-07-08

**Authors:** Bettina D. Balmer, Christian Blüthgen, Bettina Bässler, Katharina Martini, Florian A. Huber, Lisa Ruby, Amadéa Schönenberger, Thomas Frauenfelder

**Affiliations:** Institute of Diagnostic and Interventional Radiology, University Hospital Zurich, University of Zurich, CH-8091 Zurich, Switzerland; christian.bluethgen@usz.ch (C.B.); baessler_b@ukw.de (B.B.); katharina.martini@usz.ch (K.M.); florian.huber@usz.ch (F.A.H.); lisa.ruby@usz.ch (L.R.); amadea.schoenenberger@usz.ch (A.S.); thomas.trauenfelder@usz.ch (T.F.)

**Keywords:** computed tomography (CT), matrix size, 1024 × 1024 pixel, systemic sclerosis (SSc), interstitial lung disease (ILD), kernel

## Abstract

Background: Interstitial lung disease (ILD) is a frequent complication of systemic sclerosis (SSc), and its early detection and treatment may prevent deterioration of lung function. Different vendors have recently made larger image matrices available as a post-processing option for computed tomography (CT), which could facilitate the diagnosis of SSc-ILD. Therefore, the objective of this study was to assess the effect of matrix size on lung image quality in patients with SSc by comparing a 1024-pixel matrix to a standard 512-pixel matrix and applying different reconstruction kernels. Methods: Lung scans of 50 patients (mean age 54 years, range 23–85 years) with SSc were reconstructed with these two different matrix sizes, after determining the most appropriate kernel in a first step. Four observers scored the images on a five-point Likert scale regarding image quality and detectability of clinically relevant findings. Results: Among the eight tested kernels, the Br59-kernel (sharp) reached the highest score (19.48 ± 3.99), although differences did not reach statistical significance. The 1024-pixel matrix scored higher than the 512-pixel matrix HRCT overall (*p* = 0.01) and in the subcategories sharpness (*p* < 0.01), depiction of bronchiole (*p* < 0.01) and overall image impression (*p* < 0.01), and lower for the detection of ground-glass opacities (GGO) (*p* = 0.04). No significant differences were found for detection of extent of reticulations/bronchiectasis/fibrosis (*p* = 0.50) and image noise (*p* = 0.09). Conclusions: Our results show that with the use of a sharp kernel, the 1024-pixel matrix HRCT, provides a slightly better subjective image quality in terms of assessing interstitial lung changes, whereby GGO are more visible on the 512-pixel matrix. However, it remains to be answered to what extent this is related to the improved representation of the smallest structures.

## 1. Introduction

Pulmonary involvement is the second most frequent visceral complication of systemic sclerosis (SSc) [1] and the leading cause of disease-related death [2]. The British Society for Rheumatology (BSR) and British Health Professionals in Rheumatology (BHPR) state that all SSc cases should be evaluated for lung fibrosis and that treatment is to be determined by the extent, severity and the likelihood of progression to severe disease [3].

The two main types of lung involvement are pulmonary vascular disease and interstitial lung disease (ILD), the latter being the more common one [1]. Early ILD is often asymptomatic and has a high rate of false-negative results in pulmonary function tests [4]. High-resolution computed tomography (HRCT) [5] has been proven to allow early detection of lung involvement, even from subclinical stages, and has become the gold standard for the detection of ILD [2]. Nevertheless, there is so far no generally accepted expert recommendation that routine baseline thoracic HRCT should be performed in patients with SSc. Rather, the central role of HRCT in detecting SSC-ILD reflects clinicians’ dissatisfaction with other modalities [6]. Early detection and treatment of ILD may prevent deterioration of lung function [1].

SSc-ILD corresponds to non-specific interstitial pneumonia (NSIP) in most cases, whereas usual interstitial pneumonia (UIP) is encountered less frequently [7,8]. 

CT correlates of SSc-ILD are, therefore, mostly ground-glass opacity (GGO) and coarse reticulation/reticular intralobular interstitial thickening (representing underlying fibrosis) and, less frequently, honeycombing [9,10]. These pathological changes are mostly bilateral, subpleural, symmetrical and with a basal predominance [11]. Fibrosis inside focal GGO also occurs, but in contrast to, for example, lung changes in COVID-19 pneumonia, consolidations are atypical [12]. Over time, the radiographic progression involves the replacement of GGO with honeycombing/traction bronchiectasis and/or bronchiolectasis. Only about 5% of patients with GGO and nonfibrotic interstitial opacities demonstrated improvement in HRCT findings over time in a study with 41 patients [6,13].

HRCT as a post-processing feature (mainly a narrow slice thickness of 1–2 mm) is useful for lung analysis in general [14,15], and in particular for analysis of NSIP as in SSc-ILD [9,16]. It shows submillimeter structures, bronchial structures and lung parenchyma pathologies [17,18]. However, a higher spatial resolution requires a correspondingly higher processing power, more storage and faster network traffic for the resulting images [19].

On the one hand, the spatial resolution depends on the hardware. Ultra-high-resolution CT (UHRCT) has been introduced in recent years, and prior studies showed that it was helpful for better recognition of lung nodules and led to better subjective image quality in general [5,20]. However, Ultra-high-resolution CT (UHRCT) scanners are currently mainly used for cardiovascular, but not for lung imaging in the clinical setting. They lead to increased radiation exposure due to the smaller detector element size, and since the Z-axis is much shorter for smaller detectors with the same number of slices, they would require multivolume or helical acquisitions for the whole lung, which would increase the susceptibility to motion artifacts [21].

On the other hand, when using a conventional CT, spatial resolution depends on user-modifiable reconstruction parameters such as kernel and the size of the volumetric image pixel (VIP). This in turn depends on the field of view (FoV), the number of pixels of the pixel matrix and the slice thickness [22]. Using a large FoV of approximately 300–400 mm in a typical HRCT, the size of the pixel becomes relevant for a detailed spatial resolution, in order to reduce blurring or step artifacts. Different vendors have recently made larger image matrices of 1024 × 1024 (instead of 512 × 512) available as a post-processing option for CT based on the raw data. However, this double resolution per axis also requires four times the processing power and memory volume. In return, finer structures are displayed with more pixels per area. Thus, the hypothesis of our study was that a higher resolution matrix in HRCT, increases image quality and hence enables a more accurate description of typical features in SSc-ILD. 

The goals of this study were, therefore, (1) to evaluate the best kernel for HRCT to assess SSc-ILD when using a 1024 × 1024-pixel matrix and (2) to analyze the accuracy of this higher resolution matrix, compared to a standard (512 × 512-pixel matrix) HRCT for the detection and quantification of ILD in patients with SSc.

## 2. Materials and Methods

### 2.1. Study Design and Patient Characteristics

This retrospective single-institution study was approved by the institutional review board and the local ethics committee (*BLINDED*). General consent for the use of image data for scientific purposes was present for all patients. 

We retrospectively included CT scans from 50 patients, diagnosed with SSc, who underwent HRCT between May to August 2018. All patients fulfilled either the American College of Rheumatology (ACR) classification criteria or the Very Early Diagnosis of Systemic Sclerosis (VEDOSS) criteria for SSc [23,24]. No differentiation was made between patients with limited systemic sclerosis and patients with diffuse systemic sclerosis.

### 2.2. Image Acquisition

Raw CT datasets were acquired from two different scanners in our institution: a 192-slice CT scanner with a maximum spatial resolution of 0.24 × 0.24 mm^2^ (Somatom Force, Siemens Healthineers, Forchheim, Germany) and a 128-slice CT scanner with a maximum spatial resolution of 0.3 × 0.3 mm^2^ (Somatom Edge Plus, also Siemens Healthineers). The patients were scanned twice in prone position, first in full inspiration and secondly at end-expiration. 

A limited spiral CT scan was only obtained for the basal lung sections with 50 mAs, and 100 kV. In addition, two slices were acquired in the upper part using the sequence mode with tube voltage selection and automatic tube current modulation.

For kernel- assessment, a subset of 20 slices was reconstructed with an image matrix size of 1024 × 1024 pixel and with 5 different reconstruction kernels: Bv49, Bv58, Br49, Br54, Br59, Bl64, Bl57 (B = body, v = vascular, r = regular, l = lung; the higher the number, the sharper the kernel).

For the main part of the study (comparison of matrices), the entire scan was reconstructed with the two different matrix sizes 1024 × 1024 and 512 × 512 pixel using kernel Br59 (resulting from part 1). The field of view (FoV) was adjusted to the thorax diameter of the respective patient and varied between 263 and 443 mm. 

The following reconstruction parameters were equal for both scans: Iterative reconstruction algorithm: Advanced Modeled Iterative Reconstruction (ADMIRE) 3; Slice thickness 1.5 mm; increment 1 mm; lung window. 

### 2.3. Qualitative and Statistical Analysis

The image read out was performed using standard PACS software (Synedra View 18.0.0 Synedra AIM version 18 “Apollon”; Synedra Information Technologies GmbH, Innsbruck, Austria) using fixed windowing of lung.

Images were analyzed by four radiology residents with two to five years of experience in thoracic imaging. All readers were blinded to patient-related data and were presented with the images in completely randomized order regarding patient, matrix and kernel. All four readers were given a brief instruction with pictorial examples of pathologic lung changes in SSc-ILD and their definitions. A 5-point Likert scale ranging from 1 to 5 was used to assess the HRCT scans with five subcriteria for the comparison of the different kernels and six subcriteria for the comparison of the two matrices (for both, refer to Table 1). 

A total score was determined by summing up the values of each of these categories. For each patient, the variables were repeatedly collected for each kernel, respectively, for the two different matrices by four readers.

Statistical analysis was performed in R (version 4.0.3; R Foundation for Statistical Computing [25], with RStudio (version 1.4.1103; RStudio) [26]). R packages used for statistical analysis were readxl [27], tidyverse [28], ggplot2 [29], DescTools [30], WRS2 [31] multcomp [32], and irr [33].

All continuous data are given as means ± standard deviation. Categorical variables are expressed as frequencies or percentages. A two-tailed *p* value of <0.05 was considered to indicate statistical significance.

Analysis for group differences regarding kernels was performed by using robust repeated-measures analyses of variance for multilevel variables (ANOVAs) with Tukey-type comparisons for adjustment of multiple testing. 

Analysis of group differences between the two matrices was performed by using the Welch t-test for paired samples after assessing normality in the distribution of the data.

Inter-reader agreement was tested amongst all four readers using the intraclass correlation coefficient (ICC). ICCs were defined as excellent (ICC ≥ 0.75), good (ICC = 0.60–0.74), moderate (ICC = 0.40–0.59) and poor (ICC ≤ 0.39) [34]. 

Statistical significance was inferred at a *p*-value below 0.05. 

## 3. Results

### 3.1. Patient Characteristics 

The mean age was 54 years (min: 23, max: 85) and the proportion of female patients was 86% (Table 2). Most patients had a low modified Rodnan skin score (MRSS) (2.9 ± 4.7) and a good lung function (mean transfer factor for carbon monoxide or (TLCO) = 73% ± 19.6). 

Five patients (10%) had additional pulmonary diagnoses such as status post hemopneumothorax (*n* = 1), latent tuberculosis infection (*n* = 2), chronic bronchiolitis (*n* = 1), chronic obstructive pulmonary disease (COPD) stage II (*n* = 1).

### 3.2. Comparison of Different Kernels

The highest total score was reached for Br59 (19.5 ± 4.0) and the lowest for Bl64 (16.8 ± 5.5) (Table 3, Figure 1). Among the tested subcategories, the kernel Br59 achieved the highest value in “detection of pathologies” (4.0 ± 1.0; Appendix A), “depiction of bronchiole” (4.1 ± 1.0; Appendix A) and “overall image impression” (3.9 ± 0.8; Appendix A). Bl57 scored highest in sharpness (4.1 ± 0.9; Appendix A) and Bv49 highest in noise (equates to little noise) (3.6 ± 1.0; Appendix A). There was a tendency for harder kernels to score higher in sharpness and lower in noise.

There were no significant differences among the scores, with exception of the sharpness criterion between Br49 and Br59 (*p* < 0.01) and the “overall impression” criterion, between Br59 vs. Bv49 (*p* < 0.01) and Br63 vs. Bv49 (*p* < 0.01).

### 3.3. Comparison of Different Matrices

The mean overall score of the 512-pixel image matrix was significantly lower than of the 1024-pixel matrix (24.1 ± 4.1 vs. 24.7 ± 4.5, *p* < 0.01; refer to Table 4 and Figure 2). The visual score for the subcategories was significantly lower for the 512-pixel image matrix with regard to the feature “sharpness” (3.6 vs. 4.2, *p* < 0.01;Appendix A), “depiction of bronchiole (DoB)” (3.5 vs 3.7, *p* < 0.01;Appendix A) and “overall image impression (OII)” (3.7 vs. 3.9, *p* < 0.01; Appendix A) and higher with regard to extent of GGO (3.0 vs. 2.9, *p* = 0.04; Appendix A) (Figure 3) compared to the 1024-pixel image matrix. The subcategories “extent of reticulations/bronchiectasis/fibrosis” (2.99 vs. 2.94, *p* = 0.49; Appendix A) and “noise” (3.81 vs. 3.72, *p* = 0.09; Appendix A) did not show a significant difference for the two matrices. 

In summary, the 1024-pixel matrix was perceived to be sharper than the 512-pixel matrix and the smallest bronchial structures may be better displayed with it. On average, readers preferred the overall image impression of the 1024-pixel matrix. However, we could not prove that more SSC-ILD typical changes can be detected by the 1024-pixel matrix.

Figure 4 shows an example of how the same section on the same patient looks in the 512-pixel matrix (B) and in the 1024-pixel matrix (A). In a direct visual comparison, correlating with our results, it is noticeable that the 1024-pixel matrix is sharper, and the bronchiole and reticulations/fibrosis therefore better displayed. However, the GGO may be slightly better recognizable in the 512-pixel matrix.

### 3.4. Intraclass Correlation and Field of View

The calculated ICC (two way random, consistency) reflected for the comparison of the matrix good reliability [29] (The term reliability here is defined as the extent to which measurements can be replicated) (0.75–0.9) for the criteria: “Extent of reticulations/bronchiectasis/fibrosis” and “extent of GGO” (Table 5). Moderate reliability (0.5–0.75) was found for the total and poor reliability (<0.5) for sharpness, noise, “depiction of bronchiole” and “overall image impression”. For the comparison of the kernels there was found poor reliability for all these criteria.

The field of view was adjusted to the thoracic diameter of each patient. Most of the reconstructed spirals were within a field of view between 263 mm and 383 mm (89 out of 100), but the overall variance was relatively large (range 263–443 mm) (Figure 4).

## 4. Discussion

The study showed that the 1024-pixel matrix performed significantly better than the 512-pixel matrix in total and, in particular, in the subcategories of sharpness and “overall image impression”. The remaining subcategories, noise and “extent of R/B/F”, were not significantly different, but there was significantly more GGO detected when using the 512-pixel matrix.

The preliminary study did not reveal a clearly superior kernel. Very few kernel comparisons showed significant differences (namely the sharpness criterion between Br49 and Br59 (*p* < 0.01) and the « overall impression » criterion, between Br59 and Bv49 (*p* < 0.01) and between Br63 and Bv49 (*p* < 0.01).

However, there was a tendency towards Br59 being the best kernel both in most of the subcategories and in total. Unsurprisingly, only the image noise tended to be better with softer kernels (without significance). In our institute, we use the Br59 kernel among other kernels to reconstruct lung CT scans in scleroderma patients. Thus, we considered it best to use the Br59 kernel for both the 512-pixel matrix and the 1024-pixel matrix for the main study. 

To the best of our knowledge, there is no study comparing different reconstruction kernels at different matrix sizes in lung CT scans for pulmonary fibrosis.

There are various kernels used in different studies, which hampers the comparability between the study results. For example, Bartlett et al. used a soft reconstruction kernel (B46) for the 512- and 1024-pixel matrix (for energy-integrating detector-CT (EID-CT)) [35]. Higher order bronchiole were detected in the middle lobe at a higher resolution but the image noise was roughly the same in both matrices. In another study by Euler et al., on the other hand, a hard kernel (Bl64) was used for a 512- and a 1024-pixel image matrix [22]. The results showed superior image sharpness and higher image noise for the 1024-pixel matrix and no difference in the subjective image quality or the depiction of bronchial structures. Similar results are reported by Hata et al. They used a UHRCT and reported a significantly better performance of the 2048 × 2048- and the 1024 × 1024-pixel matrix than the 512 × 512 matrix in terms of overall image quality, clarity of small vessels, solid nodules, GGO, emphysema, intralobular reticulation and honeycombing but a worse performance in terms of visual noise and streak artifact [5]. 

Similar to Euler et al., we attribute this difference in the impression of image noise to the fact that if a soft kernel is used, this is the limiting factor for the resolution [22]. 

It is to be expected that the noise increases with the 1024-pixel matrix with a harder kernel just as with the 512-pixel matrix. The tendency in our study suggests that a rather hard kernel is probably advantageous nonetheless, but we cannot make a definite statement on this due to the lack of significance.

As far as we know, there are no studies about the effect of the 1024 × 1024 resolution matrix on the analysis of specific pulmonary interstitial diseases such as NSIP and UIP in SSc-ILD. Nonetheless, few studies have investigated the effect of the 1024-pixel matrix on the detectability of pathologic lung features. 

One study that investigated the influence of the higher resolution matrix on the detectability of low- and high-density lesions came to a similar conclusion that the 1024 pixel matrix offers no advantage over the 512-pixel matrix [36]. In contrast, the study by Hata et al. found an advantage of the 1024-pixel matrix over the 512-pixel matrix also in pathological findings. This difference may be because they studied cadaveric lungs with UHRCT instead of in vivo lungs with conventional CT [5]. 

In our study, the subcategory “extent of R/B/F” might have been non-significant because a relatively small proportion of scleroderma patients in our institute have relevant lung involvement. 

It is consistent with our expectations that GGO is more likely to be detected at the lower resolution matrix because it suppresses the noise in which GGO may be partially masked, just like a soft kernel (Figure 3). 

However, the definition of ground-glass opacities also needs to be looked at again in this context: according to the Fleischner Society, ground-glass refers to hazy increased lung opacity in certain areas and includes both partial filling of airspaces and interstitial thickening/increased capillary blood volume as causes [37]. If a sharper image and/or higher resolution can distinguish the two entities, the observer will possibly refer to the fine network of interstitial compaction as fibrotic change, rather than ground-glass. Thus, it seems logical that a shift from GGO to fibrotic changes/reticulations occurs when describing pathological changes. Perhaps a radiologist would eventually detect more pathologic lung changes with the 1024 matrix if he was previously well trained for the specific appearance of SSC-ILD in the 1024-pixel matrix. It might then be easier to distinguish phenomena of different etiologies resulting in similar imaging appearances (e.g., GGOs from filled airspaces vs. discrete interstitial processes).

Regarding the findings relevant for pure image impression, there was a study describing similar results: the 1024-pixel matrix offers more sharpness and partially better depiction of bronchiole than the 512-pixel matrix [35]. However, the results should be treated with caution as a rather soft kernel was used. 

In the study by Euler et al., the positive and negative characteristics of the 1024-pixel matrix balance each other out: better sharpness but also more noise (evaluated both subjectively and objectively) [22]. In contrast to our study, the FoV was kept constant, and a slightly harder kernel was used. 

### Limitations

As Euler et al. have explained in detail, if the kernel is sufficiently hard and the FOV is large enough, the VIP will be smaller with a higher matrix which results in a sharper image but also increases the image noise and the appearance of more step artifacts [22]. In our study, when mapping the clinical situation, FOV was not kept constant and, therefore, the variation of the VIP due to the different FOV was similar to the variation due to the different matrix. 

Another factor that limits the sharpness of the resulting images is the size of the apparent focal spot (Focal spot size on both- Somatom Force and Somatom Edge plus-: 0.4 × 0.5 mm^2^). The smaller the focal spot, the smaller the penumbra, hence the image also becomes sharper [38]. If we could optimize the hardware to make the focal spot smaller and keep the source-to-object distance and the object-to-receptor distance the same, we could also benefit from better software (higher image resolution).

Furthermore, we did not use a UHR scanner, unlike Hata et al. for example. The resolution was, therefore, limited by the maximum resolution of our scanners (Somatom Force: 0.24 mm, Somatom Edge Plus: 0.3 mm). However, this limiting factor was comparatively small, since only a small fraction of the 1024-spirals was reconstructed with a FOV larger than 307 mm (corresponding to a pixel size of 0.3 mm).

Most of our reconstructed CT spirals are in the range between a FOV of 263 and 383 mm, which corresponds to a pixel size of 0.51–0.75 mm for a 512-pixel matrix and a pixel size of 0.26–0.37 mm for a 1024-pixel matrix.

Another limitation is that in our study we did not have any histologic reference standard on the presence and extent of pathologic lung changes due to scleroderma. Thus, a higher score on the rubrics “extent of R/B/F” and “extent of GGO” could theoretically also represent an overdiagnosis. However, it is more likely that we have a problem of underdiagnosis of pathologic lung changes in scleroderma-related interstitial lung disease [4]. 

## 5. Conclusions

There is evidence in our study that the higher resolution matrix is more pleasing to the viewer’s eye purely in terms of the image. However, for early detection of pathologic lung lesions in scleroderma patients, our data failed to show an advantage for the 1024-pixel matrix over the 512-pixel matrix. Therefore, we consider the introduction of the 1024-pixel matrix into the clinical routine as somewhat premature, although our data show promising potential. Since the 1024-pixel matrix is sharper and allows for better visualization of the smallest lung structures, the clinical potential of the 1024-pixel matrix in the future may be to detect subtypes of GGO and to help distinguish reversible from irreversible lung injury. Regarding the kernel comparison, our data suggest that a hard lung kernel is probably beneficial, although no significant differences could be demonstrated. A larger amount of data might show significant results here.

## Figures and Tables

**Figure 1 diagnostics-12-01662-f001:**
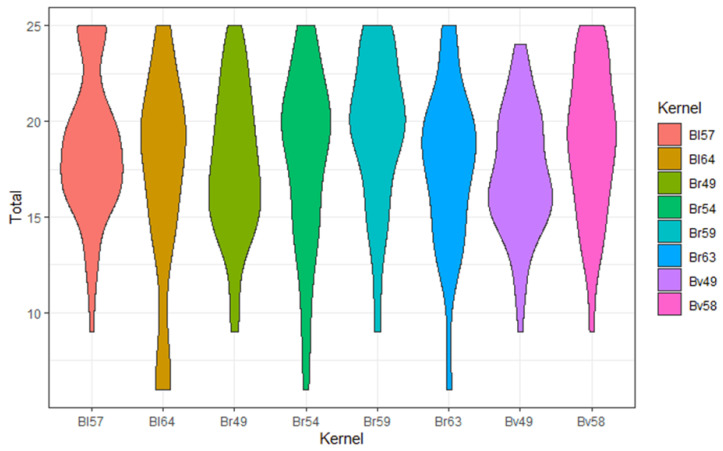
Violin plot: Total score, mean for all 4 raters, violin plot with the score distribution (X-axis: summed up 5-point Likert scale, Y-axis: Kernel) (violin plot represents kernel density estimation to show the distribution shape of the data. Wider sections of the violin plot represent a higher probability that kernel rating will take on the given value; the skinnier sections represent a lower probability).

**Figure 2 diagnostics-12-01662-f002:**
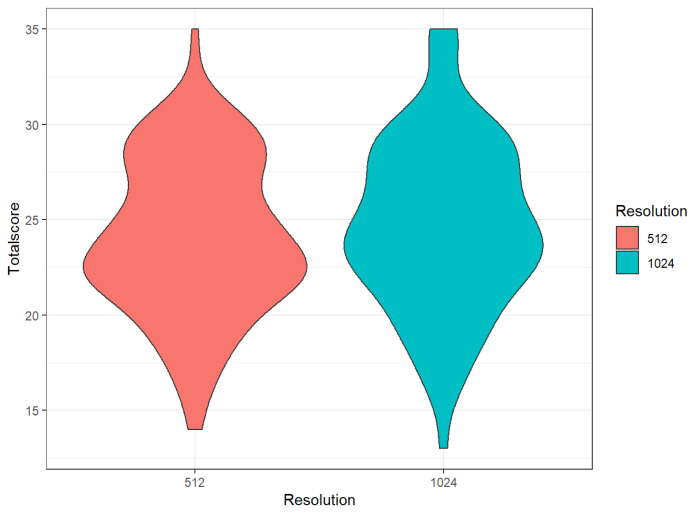
Violin plot: Total score per Matrix size (resolution) 512 and 1024 (mean for all 4 raters).

**Figure 3 diagnostics-12-01662-f003:**
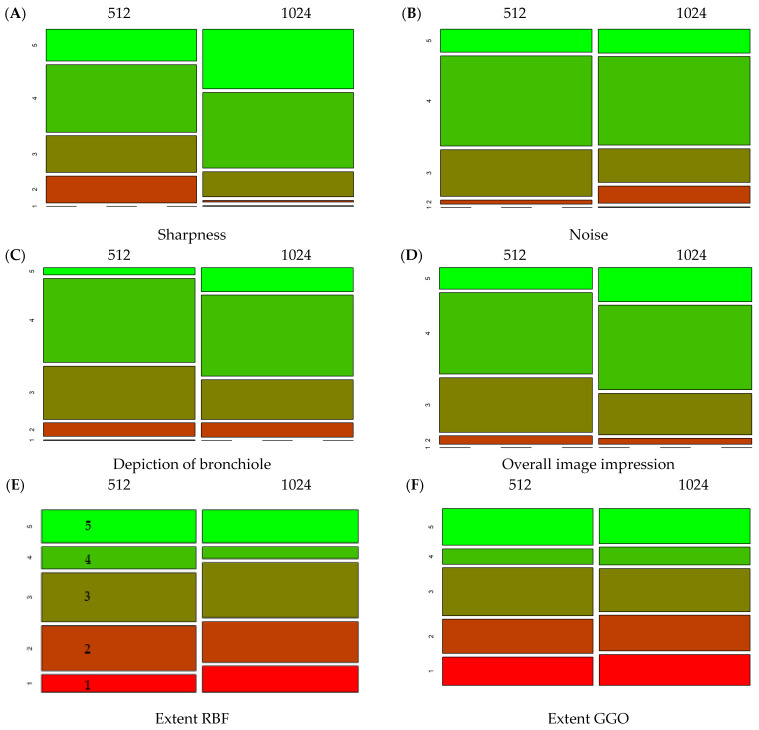
Mosaic plots with the score distribution (5-point Likert scale) of each subcategory; RBF = reticulations/bronchiectasis/fibrosis, GGO = ground-glass opacities. (**A**) shows the score distribution (1–5) for the two different matrices of the subcategory sharpness, (**B**) of the subcategory noise, (**C**) of the subcategory depiction of bronchiole, (**D**) of the subcategory overall image impression, (**E**) of the subcategory extent of reticulations/bronchiectasis/fibrosis and (**F**) of the subcategory extent of ground-glass opacities.

**Figure 4 diagnostics-12-01662-f004:**
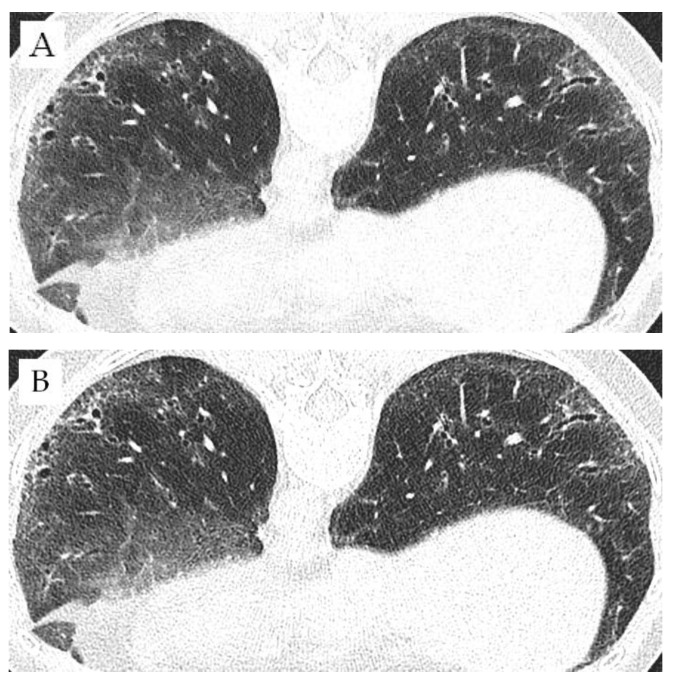
Basal lung segments of a 46-year-old patient with a modified Rodnan Skin Score (MRSS) of 7 points as a 512-pixel matrix image (**A**) and a 1024-pixel image (**B**).

**Table 1 diagnostics-12-01662-t001:** Subcriteria for the comparison of kernels and matrices, using a 5-point Likert scale.

Comparison of Kernels	Classification
Sharpness	1 = blurred, 5 = sharp
Noise	1 = noisy, 5 = clear
Detection of pathologies	1 = very bad, 5 = very good
Depiction of bronchiole	1 = very bad, 5 = very good
Overall image impression	1 = very bad, 5 = very good
Comparison of matrices	Classification
Sharpness	1 = blurred, 5 = sharp
Noise	1 = noisy, 5 = clear
Extent of reticulations/bronchiectasis/fibrosis	1 = no such pathologies, 5 = moderate, up to much of such pathologies
Extent of ground-glass opacities (GGO)	1 = no GGO, 5 = moderate up to much GGO
Depiction of bronchiole	1 = very bad, 5 =very good
Overall image impression	1 = very bad, 5 = very good

GGO = ground-glass opacities; overall image impression = the purely subjective impression of the picture quality.

**Table 2 diagnostics-12-01662-t002:** Patient characteristics.

Demography	
Number of patients	50; female: 43 (86%)
Age (in years; mean ± SD, range)	54 ± 14, 23–85
MRSS (mean ± SD, range)	2.88 ± 4.72, 0–23
Lung function	(mean ± SD)
TLC	94.87 ± 21.89
DLCO	73.15 ± 19.55
FEV1/FVC	101.93 ± 9.22
Pulmonary diagnoses	*n* (%)
St. a. HPT	1 (2)
Latent TB infection	2 (4)
Chronic bronchitis	1 (2)
COPD II	1 (2)
Smoking status	*n* (%)
Non-Smoker	39 (78)
Ex-Smoker	4 (8)
Smoker	7 (14)

SD = standard deviation, MRSS = modified Rodnan skin score, TLC = total lung capacity, DLCO = diffusing capacity of lung for carbon monoxide, FEV1 = forced expiratory volume during the first second or exhaling, FVC = forced vital capacity, St. a. HPT = state after hemopneumothorax, TB = tuberculosis, COPD = chronic obstructive pulmonary disease.

**Table 3 diagnostics-12-01662-t003:** Scores of the diverse kernels in all subcriteria and in total.

		Bl57	Bl64	Br49	Br54	Br59	Br63	Bv49	Bv58
**Sharpness**	mean	**4.08**	3.67	3.52	3.63	4.00	3.62	3.25	3.60
	sd	0.87	1.40	1.03	1.17	0.96	1.09	0.86	1.05
	*n*	60	24	60	56	60	60	60	60
**Noise**	mean	3.18	2.67	3.52	3.43	3.43	3.02	**3.58**	3.55
	sd	1	1.13	1	1.09	0.95	1.03	0.98	0.96
	*n*	60	24	60	56	60	60	60	60
**DoP**	mean	3.83	3.58	3.50	3.63	**4.03**	3.70	3.47	3.90
	sd	0.83	1.06	0.89	0.89	0.96	0.91	0.70	0.88
	*n*	60	24	60	56	60	60	60	60
**DoB**	mean	3.83	3.50	3.65	3.75	**4.12**	3.75	3.58	4.07
	sd	0.98	1.10	0.71	1.07	0.96	1.00	0.81	0.86
	*n*	60	24	60	56	60	60	60	60
**OII**	mean	3.67	3.42	3.53	3.55	**3.90**	3.50	3.38	3.82
	sd	0.82	1.18	0.83	1.03	0.84	0.89	0.74	0.81
	*n*	60	24	60	56	60	60	60	60
**Total**	mean	18.60	16.83	17.72	17.98	**19.48**	17.58	17.27	18.93
	sd	3.77	5.53	3.77	4.64	3.99	4.20	3.23	3.91
	*n*	60	24	60	56	60	60	60	60

DoP = detection of pathologies, DoB = depiction of bronchiole, OII = overall image impression.

**Table 4 diagnostics-12-01662-t004:** Score of the two matrices in the subcategories and in total.

	512		1024		*p* (Wilcox.)
	Mean	Sd	Mean	Sd	
Sharpness	3.64	0.98	**4.18**	0.76	<0.01
Noise	**3.81**	0.70	3.72	0.86	0.09
Depiction of bronchiole	3.53	0.74	**3.72**	0.83	<0.01
Overall image impression	3.70	0.76	**3.89**	0.77	<0.01
Extent RBF	**2.99**	1.38	2.94	1.40	0.49
Extent GGO	**3.05**	1.27	2.91	1.31	0.04
Total	24.08	4.09	**24.71**	4.46	<0.01

RBF = reticulations/bronchiectasis/fibrosis, GGO = ground-glass opacities.

**Table 5 diagnostics-12-01662-t005:** Interreader agreement for the assessment of the different subcategories and the total score for both, the kernel, and the matrix comparison.

Kernel		Matrix	
Criterium	ICC	Criterium	ICC
Sharpness	0.05	Sharpness	0.36
Noise	0.04	Noise	0.32
DoB	0.08	DoB	0.23
DoP	0.05	R/B/F	0.89
		GGO	0.84
OII	0.02	OII	0.29
Total	0.05	Total	0.49

ICC = intraclass correlation coefficient, DoB = Depiction of bronchiole, DoP = detection of pathologies, R/B/F = reticulations/bronchiectases/fibrosis, GGO = ground-glass opacity.

## Data Availability

Upon reasonable request to the corresponding author.

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
