# Peer review of "Influence of CT Image Matrix Size and Kernel Type on the Assessment of HRCT in Patients with SSC-ILD"

_diagnostics, 2022, doi:10.3390/diagnostics12071662_

Round 1
Reviewer 1 Report
Influence of CT image matrix size and kernel type on the assessment of 2 HRCT in patients with SSC-ILD
1) Abstract: L11-24. Interstitial lung disease (ILD) is a frequent complication of systemic sclerosis (SSc), and its early detection and treatment may prevent deterioration of lung function. Different vendors have recently made larger image matrices available as a post-processing option for computed tomography (CT), which could facilitate the diagnosis of SSc-ILD. Therefore, the objectiv of this study was to assess the effect of matrix size on lung image quality in patients with SSc by comparing a 1024-pixel matrix to a standard 512-pixel matrix and applying different reconstruction kernels. Lung scans of fifty patients (mean age 54 years, range 23-85 years) with SSc were reconstructed with these two different matrix sizes, after determining the most appropriate kernel in a first step. Four observers scored the images on a five-point likert scale regarding image quality and detectability of clinically relevant findings. Among the eight tested kernels, the Br59-kernel (sharp) reached the highest score (19.48 ± 3.99), although differences did not reach statistical significance. The 1024-pixel matrix scored higher than the 512-pixel matrix HRCT overall (p= 0.01) and in the subcategories sharpness …… Please, divide the abstract into correct sections for a research article (i.e. background, methods, results, ..).
2) 1. Introduction L36-37. Pulmonary involvement is the second most frequent visceral complication of sys- temic sclerosis (SSc) [1] and the leading cause of disease-related death [2]. Please improve this paragraph regarding SSc and add these references:
a- Systemic sclerosis: state of the art on clinical practice guidelines. RMD open, 4(Suppl 1), e000782. https://doi.org/10.1136/rmdopen-2018-000782
b- Correlations between blood perfusion and dermal thickness in different skin areas of systemic sclerosis patients. Microvascular research, 115, 28–33. https://doi.org/10.1016/j.mvr.2017.08.004
3) 1. Introduction. L 38 The two two … please correct this typo
4) 1. Introduction. L 38-43. The two two main types of lung involvement are pulmonary vascular disease and interstitial lung disease (ILD), the latter being the more common one [1]. Early ILD is often asymptomatic and has a high rate of false negative results in pulmonary function tests [3]. High-resolu tion computed tomography (HRCT) [4] has been proven to allow early detection of lung involvement, even from subclinical stages, and has become the gold standard for detection of ILD [2]. Please improve this paragraph regarding HRTC and add these references:
a- The role of chest CT in deciphering interstitial lung involvement: systemic sclerosis versus COVID-19. Rheumatology (Oxford, England), 61(4), 1600–1609. https://doi.org/10.1093/rheumatology/keab615
b- High-Resolution Computed Tomography: Lights and Shadows in Improving Care for SSc-ILD Patients. Diagnostics (Basel, Switzerland), 11(11), 1960. https://doi.org/10.3390/diagnostics11111960
5) Table 2: Patient characteristics. Could you please add the treatment regimen of all patients
6) 3. Results. Could you please underline in the text the statistical values to support the conclusions?
7) Figure 4. Basal lung segments as a 512-pixel matrix image (B) and a 1024-pixel image (A). Please ameliorate the quality of the figure.
8) 5. Conclusions L316-321. There is evidence in our study that the higher resolution matrix is more pleasing to the viewer's eye purely in terms of the image. However, for early detection of pathologic lung lesions in scleroderma patients, our data failed to show an advantage for the 1024- pixel matrix over the 512-pixel matrix. Therefore, we consider the introduction of the 1024- pixel matrix into the clinical routine as somewhat premature, although our data show a promising potential. Please underline the clinical implication of the study.
Author Response
Please see the attachement.

Reviewer 2 Report
The authors aimed to compare different post-processing changes on raw CT-data to detect ground glass opacities and fibrotic changes in data from consecutive scleroderma patients with or with our interstitial lung disease. They elaborate on the advantages of a larger image matrix 1024x1024 (instead of the more commonly used 512 x 512).
The article is clearly written, even for clinicians not confident with technical details of CT. However some questions arose when reading this article as a clinician, that might be relevant to clarify in the text:
1) To have an idea about the impact of your work I was wondering how widespread the use of HRCT is now? Do most centres with a scleroderma unit use HRCT? As I understood from the article, the HRCT is a post-processing method coming from raw conventional CT data? And do you expect UHRCT to become standard for ILD detection? Can you add in the discussion section something about the feasibility, related to irradiation/processing time /cost?
2) Can you give details on the slice thicknesses that were used for the CT in this study?
3) Please, what is meant with 'overall image impression', is it the same as 'overall image quality'? Were instructions with definitions and examples given to the observers before they scored the images? Please, add this information to the methodology.
4) I think the data that you have concerning the Kernel usage is rather showing that it does not matter which Kernel you use to visualise fibrotic or GG changes, as no significant differences in interpretation are found. Please reconsider the formulation of your conclusion on this aspect.
5) My main concern about the conclusion is that you suggest that 1024x1024 image matrix is probably better to detect ILD in scleroderma. However, you also state that ILD starts with GGO, which are scored more frequently in 512x512 image matrix. If HRCT is to be used as a diagnostic tool for ILD in scleroderma patients, it is of major interest to know whether the GGO detected on the 512x512 image matrix is related to interstitial changes later on. This would then attest the sensitivity/specificity of your data to detect ILD. It is only when this clinical correlation can be made, that we can judge which post-processing ways are best to detect ILD. Please, can you elaborate on this aspect in your discussion?
Author Response
Please see the attachement.
